# Microengineered Breast Cancer Models: Shaping the Future of Personalized Oncology

**DOI:** 10.3390/cancers17193160

**Published:** 2025-09-29

**Authors:** Tudor-Alexandru Popoiu, Anca Maria Cimpean, Florina Bojin, Simona Cerbu, Miruna-Cristiana Gug, Catalin-Alexandru Pirvu, Stelian Pantea, Adrian Neagu

**Affiliations:** 1Department of Functional Sciences, Medical Informatics and Biostatistics Discipline, “Victor Babes” University of Medicine and Pharmacy Timisoara, 300041 Timisoara, Romania; tudor.popoiu@umft.ro; 2Doctoral School, Faculty of Medicine, “Victor Babes” University of Medicine and Pharmacy Timisoara, 300041 Timisoara, Romania; miruna.gug@umft.ro; 3Department of Microscopic Morphology/Histology, “Victor Babes” University of Medicine and Pharmacy Timisoara, 300041 Timisoara, Romania; 4Center of Expertise for Rare Vascular Disease in Children, Emergency Hospital for Children Louis Turcanu, 300011 Timisoara, Romania; 5Center for Genomic Research, GENOMICA, “Victor Babes” University of Medicine and Pharmacy Timisoara, Eftimie Murgu Square No 2, 300041 Timisoara, Romania; 6Research Center for Pharmaco-Toxicological Evaluation, “Victor Babes” University of Medicine and Pharmacy, Eftimie Murgu Square No 2, 300041 Timisoara, Romania; 7Immuno-Physiology and Biotechnologies Center (CIFBIOTEH), Department of Functional Sciences, “Victor Babes” University of Medicine and Pharmacy Timisoara, No 2 Eftimie Murgu Square, 300041 Timisoara, Romania; florinabojin@umft.ro; 8Discipline of Radiology and Medical Imaging, “Victor Babes” University of Medicine and Pharmacy Timisoara, 300041 Timisoara, Romania; cerbu.simona@umft.ro; 9Department of Microscopic Morphology, Discipline of Genetics, “Victor Babes” University of Medicine and Pharmacy Timisoara, 300070 Timisoara, Romania; 10X. Department of General Surgery, “Victor Babes” University of Medicine and Pharmacy Timisoara, 300041 Timisoara, Romania; pirvu.catalin@umft.ro (C.-A.P.); pantea.stelian@umft.ro (S.P.); 11Center for Modeling Biological Systems and Data Analysis, Department of Functional Sciences, “Victor Babes” University of Medicine and Pharmacy Timisoara, No 2 Eftimie Murgu Square, 300041 Timisoara, Romania; neagu@umft.ro

**Keywords:** breast cancer, organ-on-a-chip, tumor microenvironment, microfluidics, metastasis modeling, personalized medicine, liquid biopsy

## Abstract

**Simple Summary:**

Breast cancer is one of the most common and deadly cancers affecting women worldwide. Studying this disease in controlled laboratory settings is inherently complex considering that traditional models like flat cell cultures or animal experiments cannot fully mimic how tumors grow, spread, and interact with the body. Scientists have developed a new approach called “breast cancer-on-a-chip,” which uses tiny, engineered devices to recreate real tumor conditions using human cells. The current narrative review brings together the latest advances in this technology, highlighting how it can help researchers better understand breast cancer and develop personalized treatments. We also explore how these chips are beginning to replace animal testing in drug development, offering a more realistic and human-centered approach to cancer research. These models can show how cancer behaves, how it spreads to other organs, and how it responds to treatments in a more accurate and ethical way.

**Abstract:**

**Background**: Breast cancer remains the most prevalent malignancy in women worldwide, characterized by remarkable genetic, molecular, and clinical heterogeneity. Traditional preclinical models have significantly advanced our understanding of tumor biology, yet consistently fall short in recapitulating the complexity of the human tumor microenvironment (TME), immune, and metastatic behavior. In recent years, breast cancer-on-a-chip (BCOC) have emerged as powerful microengineered systems that integrate patient-derived cells, stromal and immune components, and physiological stimuli such as perfusion, hypoxia, and acidic milieu within controlled three-dimensional microenvironments. **Aim**: To comprehensively review the BCOC development and application, encompassing fabrication materials, biological modeling of key subtypes (DCIS, luminal A, triple-negative), dynamic tumor–stroma–immune crosstalk, and organotropic metastasis to bone, liver, brain, lungs, and lymph nodes. **Methods**: We selected papers from academic trusted databases (PubMed, Web of Science, Google Scholar) by using Breast Cancer, Microfluidic System, and Breast Cancer on a Chip as the main search terms. **Results**: We critically discuss and highlight how microfluidic systems replicate essential features of disease progression—such as epithelial-to-mesenchymal transition, vascular invasion, immune evasion, and therapy resistance—with unprecedented physiological relevance. Special attention has been paid to the integration of liquid biopsy technologies within microfluidic platforms for non-invasive, real-time analysis of circulating tumor cells, cell-free nucleic acids, and exosomes. **Conclusions**: In light of regulatory momentum toward reducing animal use in drug development, BCOC platforms stand at the forefront of a new era in precision oncology. By bridging biological fidelity with engineering innovation, these systems hold immense potential to transform cancer research, therapy screening, and personalized medicine.

## 1. Introduction

### 1.1. Brief Overview of Breast Cancer Epidemiology

Breast cancer is the most frequently diagnosed malignancy and a leading cause of cancer-related death among women globally. In 2020, it accounted for approximately 2.3 million new cases and 685,000 deaths, with incidence rates steadily increasing due to population aging and lifestyle changes [1,2,3]. Mortality patterns vary by region: high-income countries show declining rates due to advances in screening and treatment, while low- and middle-income regions often face rising mortality linked to limited healthcare access [4,5].

Risk factors include non-modifiable elements—age, genetics (e.g., BRCA1/2), and family history—and modifiable ones such as alcohol and tobacco use, obesity, and hyperglycemia. These vary by region, with obesity and alcohol predominant in high-income settings, and metabolic factors more prominent in low-income countries [3,5,6].

Projections indicate a 31% global increase in cases by 2040, especially in Central sub-Saharan Africa, Oceania, and China [1,7], reinforcing the need for early detection, improved healthcare access, and risk reduction strategies [6,7].

### 1.2. From Epidemiology to the Lab: Where Are We and What Should Be Improved?

This epidemiological burden has catalyzed growing scientific interest across both clinical and preclinical research domains, driving significant investment into all stages of the disease spectrum—from early detection and screening to the extension of life expectancy, and ultimately, the pursuit of curative strategies.

Traditional in vitro and in vivo models—such as 2D monolayer cultures, static 3D systems, and animal models—have long served as foundational tools in breast cancer research. Two-dimensional (2D) cultures, in particular, have historically provided key insights into cancer cell behavior. However, they oversimplify the tumor microenvironment (TME) by lacking critical cell–cell and cell–extracellular matrix (ECM) interactions, physiological gradients, and structural organization necessary to mimic in vivo conditions [8,9,10]. These deficiencies contribute to discrepancies in drug efficacy and toxicity observed during clinical translation [11,12].

To address these shortcomings, three-dimensional (3D) models such as spheroids and organoids have emerged as more physiologically relevant systems. These cultures offer improved tissue organization and cellular heterogeneity and support the expression of lineage-specific markers when mammary epithelial cells are embedded in laminin-rich matrices like Matrigel [13,14,15]. Two-dimensional cultures fail to reveal key features such as polarity loss and invasion resulting from connexin 43 disruption, which are clearly detectable in 3D systems—demonstrating the superior ability of 3D models to capture complex breast cancer cell behaviors, although they may not necessarily preserve fidelity to the native architecture [14,16,17].

Nevertheless, conventional 3D models still fall short in capturing key organ-level features such as branching morphogenesis and dynamic biophysical stimuli. Studies have shown that spatial patterning and tissue geometry critically influence mammary gland development, which cannot be replicated using standard organoid systems [14,18]. Despite their scalability and utility in drug screening, spheroids and organoids lack the ability to reproduce mechanical cues, biochemical gradients, and tissue heterogeneity found in vivo [19,20].

Animal models—such as cell line-derived xenografts, patient-derived xenografts (PDX), and genetically engineered mice—remain indispensable for studying systemic aspects of tumor progression and metastasis [21]. However, interspecies differences in metabolism, high cost, extended timeframes, and ethical concerns limit their translational relevance and capacity to model patient-specific responses [22,23,24,25]. Moreover, murine models based on patient-derived xenografts, using as their main property immunocompromised organisms, fail in providing insight into immune–oncologic mechanisms [26,27]. A comprehensive overview of limitations seen in conventional models and advantages of “on-chip” platforms is presented in Table 1.

This review aims to examine the current state of breast cancer-on-a-chip technologies, with a focus on their development in response to the limitations of conventional models, while outlining recent technological advances, current applications, and future directions in translational cancer research and personalized medicine.

## 2. Breast Cancer-on-a-Chip Technologies: Addressing the Limitations

Breast cancer-on-a-chip (BCOC) platforms have emerged as advanced in vitro models that address key limitations of traditional systems. By integrating microfluidics, 3D cell culture, and biomimetic materials, these platforms offer physiologically relevant environments that better replicate the tumor microenvironment (TME).

First, BCOC systems reproduce critical components of the TME, including stromal cells, extracellular matrices, and vascular structures, enabling the study of complex tumor–stroma interactions [10,27]. Second, they allow fine control over microenvironmental parameters—such as oxygen gradients, shear stress, and nutrient flow—which are essential for modeling tumor dynamics and drug responses under near-physiological conditions [22,23]. Third, their compatibility with high-throughput screening makes them valuable for preclinical drug testing [10,24]. Finally, BCOC platforms can be tailored using patient-derived cells, offering a promising avenue for personalized cancer modeling [28,29]. These topics will be systematically explored in the subsequent sections of this review.

## 3. Breast Cancer Pathology

Breast cancer development and progression are driven by a complex interplay of genetic mutations, hormonal signaling, and immune modulation. These interconnected mechanisms shape tumor heterogeneity, metastatic potential, and therapy resistance, underscoring the need for advanced models that recapitulate the biological complexity of the disease.

### 3.1. Genetic Alterations and Tumor Evolution

Key driver mutations—including TP53, BRCA1/2, PIK3CA, and Estrogen Receptor 1 (ESR1)—play critical roles in tumor initiation and progression. TP53 mutations, for instance, are linked to poor prognosis and increased immune infiltration, including regulatory T-cells and M0 macrophages [30]. Mutations in ESR1 contribute to endocrine therapy resistance in estrogen receptor-positive (ER+) breast cancer and generate immunogenic neoepitopes, now being explored as targets for peptide-based immunotherapies [31,32,33].

Somatic evolution in breast cancer is also shaped by germline variants. Inherited mutations in DNA repair genes (e.g., BRCA1, CHEK2, PALB2) are associated with higher tumor mutational burden and increased sensitivity to PARP inhibitors and immunotherapies [34]. Additionally, germline-derived epitopes influence oncogene amplification and immune evasion, particularly in HER2-positive and ER-positive subtypes [35].

### 3.2. Immune System Interactions and Immune Evasion Impact on Breast Cancer Immunotherapy

The tumor immune microenvironment (TME) plays a pivotal role in determining tumor progression and response to immunotherapy. Breast cancer can escape immune surveillance via multiple mechanisms: downregulation of antigen presentation, reduced CD8+ T-cell recruitment, and accumulation of immunosuppressive cells such as Tregs and M2 macrophages [36,37,38].

In HR+/HER2− breast cancers, MAP3K1 mutations have been shown to impair antigen presentation by downregulating TAP1/2, facilitating immune escape. Emerging preclinical studies suggest postbiotics such as tyramine may restore MHC-I expression, enhancing immunotherapy response [39].

One of the most promising treatments for cancers of the mammary glands is immunotherapy, although the tissue and cellular targets of these treatments are varied and debatable. Stromal immune microenvironment is highly heterogeneous, consisting of Tumor-Infiltrating Lymphocytes (TILs) and well-organized immune cells as Tertiary Lymphoid Structures (TLSs). Within the same BC molecular subtype, significant differences between TLS+ and TLS− subgroups were found to significantly affect BC recurrence, lympho-vascular invasion (LVI), and perineural invasion (PNI). BC-associated TLS appears to have a local protective function against tumor spread by encouraging rapid blood vessel maturation, which is followed by a reduction in LVI or PNI and recurrence. Subgroups TNBC/TLS− and HER2+/TLS− (HER2 and luminal B_HER2) are found to be extremely aggressive, with a high recurrence rate that appears to result from a crosstalk between stromal and tumor components. The evaluation of cellular pathways influencing vessel proliferation and maturation is necessary to confirm the effect of TLS on stromal angiogenesis [40].

### 3.3. Hormonal Signaling and Tumor–Immune Crosstalk

Recent work has detailed the dual roles of immune cell populations within the breast tumor microenvironment (TME), emphasizing both their tumor-suppressive and tumor-promoting functions. Key cell types—including T-cells, natural killer (NK) cells, myeloid-derived cells, innate lymphoid cells, mast cells, and eosinophils—contribute to this balance through mechanisms such as cytokine signaling, cell–cell interactions, and the exchange of extracellular vesicles. Moreover, emerging therapeutic strategies aim to reverse the immunosuppressive nature of breast TME. These include chimeric antigen receptor (CAR)-engineered T- and NK cells, cancer vaccines, therapies that induce immunogenic cell death, epigenetic modulators (e.g., DNMT and HDAC inhibitors), cytokine-based interventions, and other novel immunotherapy approaches [41].

Hormonal regulation, particularly via ER signaling, is a major driver of tumor growth in ER+ breast cancer. Mutations in ESR1 confer ligand-independent receptor activity, supporting tumor growth despite endocrine therapy [42,43]. These tumors often exhibit a TME enriched in programmed death-ligand 1 (PD-L1+) macrophages and Tregs, highlighting potential for combination endocrine–immunotherapy strategies [44].

### 3.4. Implications for Therapeutic Targeting

Understanding the cellular, genomic, and immunologic heterogeneity of breast cancer tumor–stroma interaction has significant therapeutic implications. The integration of germline and somatic data enables genomics-guided treatment strategies, particularly in DNA Damage Repair (DDR)-deficient tumors. Vaccines targeting ESR1 neoantigens, inhibitors of immune checkpoints, and modulation of the tumor stroma are all emerging approaches aimed at overcoming resistance and enhancing therapeutic response [32,45].

## 4. Fabrication of Microfluidic Platforms

### 4.1. Fabrication Materials

Among the various materials available for platform fabrication, polydimethylsiloxane (PDMS) stands out as the most employed, being a silicon-based organic polymer typically structured via soft lithography techniques. This approach enables the creation of organ-on-a-chip (OoC) systems that are flexible, biocompatible, and optically transparent [46]. However, despite its favorable properties, PDMS presents certain limitations. Its high permeability for small hydrophobic molecules and gases can hinder accurate assessment of drug concentrations and complicate the creation of anaerobic conditions. Additionally, soft lithography is a relatively low-efficiency method that poses challenges for scalability, thereby limiting the transition from laboratory prototypes to high-throughput industrial production [27,47].

Alternatives to PDMS are of high interest in the OoC industry, thus multiple formulas have been tested. Elastomeric synthetic materials such as polyester toner have shown promise due to their low drug absorption, good elasticity, and biocompatibility. Polyester-based vein-on-chip systems have demonstrated enhanced endothelial cell adhesion and proliferation when treated with fibronectin and oxygen plasma [48,49]. Another emerging material is poly(lactic-co-glycolic acid) (PLGA), previously used in drug delivery, now adopted in lung-on-chip models as a porous nanofiber scaffold with tunable thickness for 3D co-culture of lung cancer cells and fibroblasts [50]. Similarly, poly(octamethylene maleate (anhydride) citrate) (POMaC), a biodegradable elastomer, has mechanical properties that match soft biological tissues and has been employed in chip fabrication [51].

Thermoplastics such as poly(methyl methacrylate) (PMMA) provide high mechanical strength, low permeability to small molecules, and recyclability [52,53]. CO_2_ laser engraving of PMMA provides great precision and micro-patterns [27]. A recently validated approach in breast cancer-on-chip models involves using PMMA as the primary substrate for device fabrication, integrated with an electrospun polylactic acid (PLA) scaffold. This scaffold structurally partitions the culture chamber into upper and lower compartments, providing a biomimetic environment conducive to cell growth. PMMA is emerging as a promising alternative to traditional polydimethylsiloxane (PDMS) in tumor-on-chip platforms due to its superior mechanical strength, biocompatibility, low permeability to small molecules, and high optical clarity [27].

Polyetheretherketone (PEEK), though limited by low degradability, has gained attention for its affordability and stability. When blended with polyglycolic acid (PGA), the resulting PEEK/PGA scaffolds show improved degradation and biocompatibility, making them suitable for microfluidic applications [54,55,56].

Natural biomaterials also present valuable alternatives. Collagen and elastin-based membranes have been used to fabricate biodegradable lung-on-chip systems that avoid the small-molecule absorption seen with PDMS and offer tunable mechanical and structural properties [57]. Type I collagen has served as a stable gel matrix in cardiac micro-bioreactors mimicking myocardial bundles, while fibrin–collagen gels have supported vascularized 3D spheroids [56,58]. Furthermore, gelatin–methacryloyl (GelMA) hydrogels combine biological adhesion with photo-crosslinking stability, supporting long-term cultures such as liver-on-chip models for up to 30 days [59].

Biomaterials used in organ-on-a-chip (OoAC) platforms are broadly categorized into natural and synthetic polymers. Natural polymers, derived from sources such as the extracellular matrix, plants, or animal tissues, are highly biocompatible and widely used in applications like tissue engineering and drug delivery due to their ability to support cell adhesion, proliferation, and mechanical stimulation [56]. However, their use is limited by batch-to-batch variability and the need for careful purification and sterilization. In contrast, synthetic polymers offer precise control over physical and chemical properties and exhibit greater consistency. Yet, they often lack inherent bioactivity and require chemical modification to support cellular functions. The choice between natural and synthetic materials depends on the specific requirements of the tissue model being replicated in the OoAC system [56,60].

**Collagen**, a widely utilized natural polymer, is valued for its excellent biocompatibility, enzymatic degradability, and intrinsic cell-adhesion motifs [61]. However, its poor mechanical strength and limited structural stability in hydrated conditions present challenges, although these can be addressed through mechanical tuning. Standard sterilization methods like ethylene oxide and gamma irradiation may denature the protein structure [62,63]. Collagen is commonly applied as a coating to enhance cell attachment and has been successfully employed in gut-on-a-chip, bone-on-a-chip and breast cancer-on-a-chip-models [62,64,65].

**Fibrin** offers a bioactive, biodegradable scaffold suitable for cell encapsulation. While mechanically weak, it can be reinforced through crosslinking and is applied in lung, vascular, and muscle models [61,66], with usage in breast cancer microenvironments, usually in combination with **fibronectin** and **hyaluronic acid** [67], which in turn have major advantages in cell signaling and barrier modeling [68]. Moreover, increased concentration of fibronectin induced overexpression of matrix metalloproteinases secreted by fibroblasts, which in turn stimulated metastatic cell migration of Triple-Negative Breast Cancer (TNBC) [69].

Although less commonly applied in breast cancer modeling, **gelatin, chitosan, and alginate** are frequently used in vascular models owing to their structural stability and tunable stiffness, with chitosan possessing the additional advantage of antimicrobial activity [56,61,70,71].

### 4.2. Fabrication Techniques

Microfluidic chip fabrication involves diverse techniques, each with unique advantages and limitations. Soft lithography, using materials like PDMS or polystyrene, offers cost-effective, high-throughput production of intricate microstructures without requiring a cleanroom, though it needs a prefabricated stamp master [72]. Photolithography, used on flat substrates like glass or PDMS, enables high-resolution patterning across various materials but requires cleanroom conditions and substantial initial investment [73]. Injection molding enables large-scale production with consistent quality using thermoplastics such as PMMA or cyclic olefin copolymer (COC), but it comes with high initial costs and limited design flexibility [74]. Hot embossing, suitable for materials like polylactic acid (PLA) or polycarbonate (PC), enables efficient replication with structural precision, yet requires careful parameter control and thermoplastic use [75]. Etching techniques (e.g., with p-HEMA or silk fibroin) provide high precision and rapid prototyping but involve costly equipment and hazardous chemicals [76]. Laser cutting, often used with paper, offers excellent accuracy and quick development, though it produces toxic by-products and demands post-processing [77]. Three-dimensional printing supports rapid, flexible prototyping with a wide material range (e.g., PEG, collagen, alginate), though post-processing and size limitations persist [56,78].

Electrospinning has emerged as a versatile and cost-effective technique for engineering nanofibrous scaffolds tailored to breast cancer (BC) research and therapy. By mimicking the topography and architecture of the extracellular matrix, electrospun materials provide a biomimetic environment conducive to 3D cell culture, drug delivery, and gene therapy applications [79]. These fibrous constructs, with tunable mechanical properties and fiber diameters ranging from micrometers to nanometers, enable enhanced drug loading and controlled release—helping to overcome key limitations of conventional chemotherapy, including poor bioavailability, off-target toxicity, and insufficient tumor accumulation. Recent advancements in electrospinning techniques have highlighted their value not only in BC drug delivery systems but also as in vitro models for diagnosis and therapeutic screening, thereby supporting their integration into BCOC platforms for more personalized and effective treatment strategies [47,80]. As shown in Table 2, each method presents trade-offs between resolution, scalability, and cost, influencing the choice of materials for specific applications.

Despite remarkable progress, the fabrication of breast cancer-on-a-chip devices remains technically demanding, resource-intensive, and limited by the current inability to reproduce physiologically relevant vascular networks and real-time biosensing without perturbing the microenvironment [28]. Future research should be directed towards developing standardized, cost-effective fabrication techniques, advancing microvascular engineering for perfusable and physiologically scaled networks, and integrating non-invasive, real-time biosensors that can monitor dynamic cellular and molecular events without disrupting the tumor microenvironment.

## 5. The Role of the Tumor Microenvironment in Breast Cancer Progression

The tumor microenvironment (TME) is a dynamic and integral component of breast cancer biology, critically shaping tumor initiation, progression, immune evasion, angiogenesis, and metastasis. It comprises a complex network of cellular (e.g., tumor cells, immune cells, fibroblasts), biochemical, and biomechanical signals that collectively influence cancer cell behavior.

A landmark study by Maffini et al. demonstrated that exposure of mammary epithelial cells to the carcinogen N-nitroso-N-methylurea (NMU) alone did not induce neoplastic transformation in vitro. However, tumorigenesis occurred when the same cells were placed within NMU-exposed stroma, emphasizing the TME’s instructive role in initiating malignancy [81].

Breast cancer TME is represented by a multitude of cell types such as fibroblasts, myoepithelial cells, adipocytes, immune system-related cells—macrophages, T-cells, and NK cells—and stromal cells bound to a complex matrix.

**Fibroblasts** are typically quiescent but become activated during mammary gland involution, a state shown to promote tumorigenesis. In cancer, these cells are known as cancer-associated fibroblasts (CAFs) [82]. CAFs secrete growth factors such as chitinase-3-like-1 (Chi3L1), activating MAPK and PI3K pathways to promote tumor proliferation and immune suppression through impaired T-cell recruitment [14,83].

**Breast adipocytes**, alongside immune cells, support glandular tissue and influence the tumor microenvironment (TME). Their phenotype shifts based on factors like breast density, menopausal status, and Body Mass Index (BMI). In the TME, they show reduced lipid content, secrete pro-inflammatory cytokines (e.g., IL-6, TNF-α, VEGF), and express extracellular matrix (ECM)-degrading proteases, all of which promote CAF activation, invasion, and metastasis. Obesity contributes to chronic inflammation, fueling tumor-supportive signaling and fatty acid metabolism dysregulation, which tumors exploit for growth. Tumor-associated adipocytes also overexpress tumor-promoting factors like versican, enhancing angiogenesis [9,84,85]. **Adipose-derived stromal cells (ASCs)**, located around adipocytes, can differentiate into various mesenchymal lineages. In obesity, ASCs lose this ability, adopting a myofibroblast-like phenotype that may accelerate tumor progression. This phenotypic shift is likely driven by chronic inflammation. Importantly, BMI may affect therapeutic outcomes and should be considered in personalized medicine [9,86,87]. Coculturing adipose tissue on microfluidic models of BC is nearly mandatory for understanding TME involvement. A novel four-channel microfluidic device was developed to dynamically co-culture triple-negative MDA-MB-231 breast cancer cells and adipose-derived stem cells (ASCs). The platform features a PDMS layer with four channels and a bottom agarose layer enclosed in a Plexiglas chamber, enabling passive diffusion of nutrients and signaling molecules without shear stress, supporting real-time intercellular communication. Using this device, researchers observed that co-cultured cancer cells showed increased growth, invasive morphology, and directional polarization toward ASCs. Notably, these cancer cells also developed greater resistance to paclitaxel compared to monocultures [88].

In an attempt to enhance the diversity of breast cancer–adipose tissue preclinical models, Hamel et al. suggest three key strategies: (1) sourcing matched mammary adipose tissue and tumor samples from a diverse donor population; (2) thoroughly characterizing the breast tumor microenvironment (TME) across various ages, ethnicities, and comorbidities; and (3) increasing cellular complexity by including endothelial and immune cells [89].

Myoepithelial cells, situated between the terminal ducts, lobules, surrounding stroma, and adipose tissue, play a pivotal role in maintaining breast tissue integrity. These specialized cells contribute to the formation of the basement membrane (BM), providing both structural support and tumor-suppressive functions by physically restraining epithelial cell dissemination and secreting anti-invasive molecules such as nexin II, α1-antitrypsin, MMP-1 inhibitors, and thrombospondin-1 [90,91]. Thus, myoepithelial cells are recognized as natural barriers to tumor progression. However, under the influence of the tumor microenvironment (TME), they may acquire genetic and epigenetic alterations, leading to functional dysregulation and undermining their suppressive role. Overexpression of proteolytic enzymes (e.g., MMPs, serine proteases, cathepsins) by tumor cells that degrade the BM and/or intrinsic alterations in the myoepithelial cells themselves, leading to secretion of factors that modify the local microenvironment and weaken barrier integrity, are two suggested theories of the barrier evasion strategy of the BC [92,93,94]. Given the dynamic hormonal responsiveness of breast tissue and the cellular specificity of tumor origin, it is essential to incorporate the TME—particularly the myoepithelial layer—into advanced in vitro models [9].

Innate and adaptive immune cells—including macrophages, dendritic cells, NK cells, myeloid-derived suppressor cells (MDSCs), and T lymphocytes—play dual roles in either suppressing or promoting tumor progression [95]. In breast cancer (BC), the extent and type of immune infiltration vary depending on tumor subtype, particularly in relation to hormone receptor expression (ER+, HER2+, and TNBC) [96]. Among immune populations, tumor-associated macrophages (TAMs) are the most prevalent, polarizing into either pro-inflammatory M1 or anti-inflammatory M2 phenotypes. M1 macrophages, activated by cytokines such as interferon-γ, exert anti-tumor functions. In contrast, M2 macrophages, stimulated by regulatory cytokines like TGF-β1 and interleukins IL-4, IL-10, and IL-13, support tumor progression through immunosuppression, angiogenesis, and tissue remodeling, with observed predominance of M2-like TAMs in the BC tumor microenvironment correlating with poor clinical outcomes, including larger tumor size, higher grade, estrogen receptor (ER) negativity, and reduced overall survival [9,97,98]. In addition to M2-like phenotype polarization, immune evasion is seen also through MDSCs’ inhibition of T-cell activation via arginase and reactive oxygen species. Together, these contribute to immune suppression through the expression of checkpoint molecules such as PD-L1 and the recruitment of regulatory T-cells [99,100]. On the other hand, functional analyses made on a BC model on a chip revealed that, while macrophages alone enhanced cancer cell proliferation, migration, and invasion speed, the co-presence of T-cells (Tri condition) significantly suppressed these behaviors. This shift was accompanied by elevated levels of RANTES and leptin, cytokines known for their dual roles in tumor biology. Notably, leptin was linked to increased polarization of macrophages toward the M1 (anti-tumor) phenotype, as confirmed by CD86 expression via flow cytometry [101].

The breast **extracellular matrix (ECM)**, produced by fibroblasts and endothelial cells, includes type I-IV collagens, laminins (LM-111, LM-332), fibronectin, and hyaluronan. It supports cell structure and regulates processes like adhesion, migration, and proliferation, being able to modulate tumor progression and therapy response [9]. Collagen plays a crucial role in tumor progression by regulating cancer cell alignment and migration, as demonstrated on microfluidic platforms. Human breast cancer cells cultured within 3D-printed collagen scaffolds on such platforms aligned along the direction of collagen fibers, influencing tumor development [102]. Similarly, aligned collagen bundles—designed to mimic tumor-associated stromal architecture—guided cancer cell movement through contact cues, promoting directional migration [103]. By influencing the diffusion of nutrients, oxygen, and therapeutic agents, the ECM can act as a barrier to effective drug delivery [104]. Cell-free patient-derived scaffolds (PDSs) were repopulated with breast cancer cell lines to monitor how individual microenvironments influence cancer behavior. Gene expression analysis revealed scaffold-induced changes in epithelial–mesenchymal transition and pluripotency markers, which correlated with clinical and prognostic features—especially when estrogen receptor (ERα) status between the cell line and scaffold matched, demonstrating that PDSs can uncover malignancy-promoting cues within the cancer niche, offering a promising complementary tool for breast cancer diagnosis and stratification [105].

In terms of **angiogenesis**, hypoxic conditions in the TME lead to HIF-1α stabilization and increased VEGF production, which drives endothelial proliferation and vessel formation [106,107]. TAMs and CAFs further enhance angiogenesis by secreting pro-angiogenic cytokines and remodeling the ECM [108]. Patient-derived BC tumoroids on chip microvasculature replicate essential features of stromal and vascular dysfunction, exhibiting increased deposition of hyaluronic acid and collagen, degradation of the vascular glycocalyx, reduced perfusion, and elevated interstitial fluid pressure—all of which contribute to impaired drug delivery to tumor cells [109].

Acidity is a hallmark of the tumor microenvironment, primarily resulting from the accumulation of metabolic byproducts such as hydrogen ions, lactic acid, and pyruvic acid produced during aerobic glycolysis—a key energy source for cancer cells [110]. This metabolic activity leads to a gradual decrease in local pH from physiological levels (~7.4) to more acidic values (~6.5–6.8), which contributes to immune evasion by impairing leukocyte function and enhances tumor aggressiveness by promoting cell migration and drug resistance through reduced drug uptake [111,112]. Conversely, this acidic milieu offers opportunities for targeted drug delivery, as it can trigger the release of pH-sensitive nanoparticles [112,113].

To replicate these acidic conditions in cancer-on-a-chip systems, researchers employ strategies such as the direct addition of lactic acid to culture media or mimic diffusion limitations found in dense tumor tissues. For instance, Yan et al. [113,114,115] developed a multilayered scaffold system that supports high-density breast cancer cell growth, effectively creating a metabolite-rich, acidic microenvironment sustained for up to one week. This approach enables more accurate modeling of tumor immunosuppression, chemoresistance, and pH-responsive drug release.

Hypoxia is widely recognized as a central feature of solid tumor physiology, going hand in hand with it, and generating low pH, contributing to therapeutic resistance by limiting drug delivery, reducing proliferation, and promoting aggressive phenotypes through gene expression changes [114,116]. Cancer-on-a-chip platforms replicate hypoxia via two main strategies: atmospheric control and chemical oxygen depletion. Devices using polycarbonate or PMMA materials regulate oxygen diffusion by perfusing nitrogen or pre-mixed gases, achieving O_2_ levels as low as 1.8%. Alternatively, oxygen scavengers like Na_2_SO_3_ or pyrogallol deplete oxygen within microchannels [117,118,119,120]. Genetic models overexpressing hypoxia-inducible factors offer additional biochemical mimicry [116]. These systems enable drug screening for hypoxia-targeted therapies in a physiologically relevant context [113], while characterizing the lower potency that some drugs have in hypoxic microenvironment conditions [119].

Raising the acidic pH of breast cancer TME through oral intake of Na_2_CO_3_ was shown to lower its extravasation and capacity for metastasis in mice, possibly via reducing the activity of the matrix-remodeling protease cathepsin B [121]. However, clinical translation was hardly feasible due to the high risk of iatrogenic alkalosis; thus, CaCO_3_ nanoparticles were tested on a breast-cancer-on-a-chip model as novel molecules that should not influence systemic pH. Persistent buffering of the chip’s chambers with nanoCaCO_3_ maintained an average pH of 7.25 =/−0.07, while inhibiting both tumor growth and metastasis [122].

In metastasis, the TME facilitates epithelial-to-mesenchymal transition (EMT) through TGF chip signaling, ECM degradation via MMPs, and intravasation through angiogenic vessels [108,123]. At secondary sites, tumor-derived factors such as Vascular Endothelial Growth Factor (VEGF) and Placenta Growth Factor (PlGF) help establish pre-metastatic niches that promote immune tolerance and metastatic colonization [124].

Owing to this complexity, conventional in vitro models fail to capture the intricate interactions of the tumor microenvironment. To address this, 3D biomimetic-on-chip platforms have been developed to reproduce tumor–stroma crosstalk, extracellular matrix remodeling, and biochemical gradients within controlled microenvironments [125,126].

## 6. Microfluidic Modeling of Particular Subtypes

Breast cancer (BC) is a highly heterogeneous disease that requires precise classification to inform effective treatment strategies [127]. Variability arises from differences in tumor origin, invasiveness, histological grade, lymph node involvement, and biomarker expression. Molecular classification based on immunohistochemical detection of estrogen receptor (ER), progesterone receptor (PR), and human epidermal growth factor receptor 2 (HER2) status divides BC into four main subtypes [128]:**Luminal A**: The most common subtype; ER+/PR+, HER2−, low proliferation; generally, has a favorable prognosis and responds well to endocrine therapies [129].**Luminal B**: ER+, variable PR and HER2 expression; more aggressive and less responsive to hormone therapy.**HER2-enriched**: Defined by HER2 overexpression; typically, high-grade and associated with poorer outcomes [130].**Triple-negative breast cancer (TNBC)**: Lacks ER, PR, and HER2 expression; often linked to BRCA1/2 mutations; associated with high proliferation, aggressiveness, and poor prognosis.

TNBC itself is molecularly diverse and subdivided into molecular subtypes **basal-like 1 and 2** (~75% of TNBCs), sharing basal gene signatures but differing in immune response profiles, **Luminal Androgen Receptor (LAR)**, being characterized by androgen receptor signaling, and **mesenchymal (M)** or **mesenchymal stem-like (MSL)**, both showing epithelial–mesenchymal transition (EMT)-related gene expression [9,131].

Given the distinct molecular profiles and clinical behaviors of breast cancer subtypes, breast cancer-on-a-chip models have been increasingly tailored to reflect this heterogeneity—most notably focusing on the more prevalent luminal A and the clinically aggressive triple-negative subtypes.

### 6.1. Breast Cancer-on-a-Chip Models of Ductal Carcinoma in Situ

Due to its early, non-invasive nature and confinement within the mammary ducts, ductal carcinoma in situ (DCIS) is often modeled independently using microfluidic systems designed to replicate ductal structure and study its progression into invasive breast cancer.

A landmark study by Bischel et al. introduced a hydrogel-based mammary duct model in which MCF10A epithelial cells formed a polarized monolayer lining a lumen embedded within a fibroblast-rich extracellular matrix (ECM). Upon introducing MCF10.DCIS cells into the lumen, paracrine signaling from surrounding fibroblasts alone was sufficient to initiate invasive behavior [132].

Microfluidic DCIS models have also proven valuable for therapeutic testing. Ayuso et al. designed a PDMS rod-based platform to create parallel mammary ducts embedded in fibroblast-containing ECM, allowing for dynamic perfusion of media and drugs. This system was used to simulate metabolic stress conditions such as hypoxia, nutrient deprivation, and oxidative stress—all of which were shown to enhance tumor aggressiveness [133]. Additionally, it facilitated testing of tirapazamine (TPZ), a hypoxia-activated prodrug that selectively targeted the hypoxic tumor core while sparing surrounding epithelial and stromal cells [133].

Similarly, Choi et al. developed a compartmentalized microdevice in which fibroblasts cultured in a lower chamber influenced MCF10-DCIS spheroids located in an upper compartment. When perfused with paclitaxel, the system showed only a moderate reduction in spheroid growth, highlighting the impact of drug delivery dynamics and microenvironmental complexity on therapeutic efficacy [134].

Despite these advancements, current models often lack key components of the native TME—such as myoepithelial cells, adipocytes, immune populations, and vascular structures. Incorporating these elements remains a challenge but would greatly enhance biological relevance. Nevertheless, microfluidic platforms continue to offer powerful, tunable environments for studying DCIS progression and testing therapeutic strategies in a spatially and temporally resolved manner.

### 6.2. Breast Cancer-on-a-Chip Models of Luminal a Subtype

Luminal A breast cancer, characterized by high estrogen receptor (ER) expression, progesterone receptor positivity (PR ≥ 20%), human epidermal growth factor receptor 2 (HER2) negativity, and low proliferative index (Ki-67 < 14%), represents the most prevalent and prognostically favorable molecular subtype of breast cancer [127]. To model this subtype with greater physiological accuracy, microfluidic tumor-on-a-chip systems have increasingly incorporated luminal A-specific cell lines, particularly **MCF-7** and **T47D**, to recreate key features such as epithelial–stromal interactions, extracellular matrix (ECM) remodeling, vascular integration, and immune involvement within controlled three-dimensional (3D) microenvironments.

Nashimoto et al. developed a **vascularized microfluidic tumor model** by co-culturing MCF-7 cells with human lung fibroblasts and human umbilical vein endothelial cells (HUVECs). Angiogenic sprouts formed perusable lumens that supported long-term tissue viability and allowed for real-time studies of drug delivery within a vascularized luminal A tumor context [135]. Subsequently, by using a 3D perfused biomimetic microfluidic platform, our team validated the co-culture of MCF-7 cell lines with human bone marrow derived mesenchymal stem cells (hBM-MSCs) [135]. We proved the hBM-MSCs’ ability to differentiate into endothelial cells when they interact with MCF-7 cells in a perfused microfluidic model by applying an in situ unique immunofluorescence (IF) technique, consisting of IF reagent injections into the tumor and vascular compartments of a microfluidic device, followed by confocal microscopy assessment. A VE-cadherin-positive endothelial phenotype was acquired by hBM-MSCs lining microvascular channels, which then continuously covered the microchannels as an endothelium-like layer. Subsequently, network formation assay provided by IKOSA analysis platform enabled us to use the network formation assay for in vitro angiogenesis assessment to identify and assess differentiated hBM-MSC branching points, loop presence, and cell coverage. All these important parameters for assessing the hBM-MSCs’ capacity to differentiate into endothelial cells and their capacity to connect with one another to mimic the endothelial layer were automatically calculated by the program. MCF-7 cells infused into the tumor compartment maintained E-cadherin expression throughout the whole experiment and transmigrated into the microvascular compartment (Figure 1) [135].

To further examine **stromal remodeling and ECM dynamics**, Gioiella et al. constructed a compartmentalized chip where MCF-7 cells were co-cultured with fibroblast-assembled ECM. This model revealed tumor-induced stromal activation, including overexpression of fibronectin and hyaluronic acid, and demonstrated significant collagen remodeling, paralleling in vivo observations in luminal A tumors [20].

ECM composition was systematically interrogated by Montanez-Sauri et al., who embedded T47D luminal A cells in various ECM combinations using a high-throughput microfluidic screening platform. Both monocultures and co-cultures with human mammary fibroblasts exhibited distinct morphological and proliferative responses depending on the matrix, with fibronectin- and laminin-rich environments promoting subtype-specific behaviors—highlighting the central role of microenvironmental cues [136].

Immune–tumor interactions have also been explored using microfluidic designs. Aung et al. incorporated MCF-7 spheroids, monocytes, and endothelial cells into a **photopatterned gelatin methacrylate hydrogel**, demonstrating that hypoxic tumor conditions enhanced T-cell recruitment and infiltration. Monocyte co-culture further amplified this immune response, offering insights into immune dynamics within the luminal A microenvironment [137].

On the therapeutic front, Dereli-Korkut et al. engineered a **three-layer PDMS microfluidic array** that cultured MCF-7 cells in 3D ECM adjacent to endothelial-lined channels, effectively simulating vascular flow. This configuration enabled real-time monitoring of drug-induced apoptosis and served as a functional platform for patient-specific therapeutic screening [138].

Complementing these efforts, Khoo et al. evaluated the combined effect of low-dose **aspirin and doxorubicin** on a microfluidic model of circulating breast tumor cells. Co-treatment significantly improved apoptosis and tumor cell eradication, particularly in ER+, PR−, and HER2− (luminal A-like) subtypes. Over seven days, the treatment also reduced cancer stem-like cell populations and impaired colony formation, likely due to anti-inflammatory effects associated with reduced IL-6 levels [139].

### 6.3. Breast Cancer-on-a-Chip Models of Triple-Negative Breast Cancer (TNBC) Subtype

**TNBC** represents a clinically defined but biologically diverse group of malignancies that differ markedly in their histological features, genomic landscapes, and immunological characteristics. Despite their shared lack of estrogen receptor (ER), progesterone receptor (PR), and HER2 expression, TNBCs encompass multiple distinct disease subtypes. Advances in high-throughput sequencing and integrative omics approaches have highlighted the extent of this heterogeneity and provided deeper insights into the molecular mechanisms driving TNBC pathogenesis, underscoring the challenges it presents for effective therapeutic targeting [140].

Insert-like and tumor-on-chip (ToC) systems were used to replicate an in vivo study testing olaparib (a PARP inhibitor) with dinaciclib (a CDK inhibitor) in TNBC. The combination reduced tumor growth in models with high Sam68 and RAD51 expression, key mediators of PARP inhibitor resistance, supporting the relevance of ToC models for TNBC drug testing. These findings highlight the potential of combining olaparib and dinaciclib as an effective therapeutic approach for aggressive breast cancer [27,141,142].

L-TumorChip is a modular three-layered microfluidic platform designed for high-throughput drug screening. Integrating vascular and stromal compartments, it captures key features of TNBC biology, including cancer cell invasion through leaky vasculature and angiogenesis. Importantly, it allows investigation of stromal influence: while normal fibroblasts enhance apoptotic responses to doxorubicin, cancer-associated fibroblasts (CAFs) significantly delay drug pharmacokinetics, highlighting the TME’s role in therapy response [143].

The role of vascular crosstalk was explored by Shirure et al. as well, who engineered a microfluidic system mimicking early-stage tumors interacting with the arterial end of capillaries. They demonstrated that TNBC-derived proangiogenic factors diffused against interstitial flow, inducing sprouting angiogenesis from arterial capillaries—emphasizing how tumor-secreted signals guide neovascularization under physiologically relevant flow conditions [144].

Devadas et al. employed a two-lumen microfluidic platform to study bidirectional migration between epithelial and endothelial cells using a panel of breast cancer cell lines. Notably, MDA-MB-231 (TNBC) cells displayed significantly greater motility than MCF-7 (luminal A) or MCF10A (non-tumorigenic) lines, underlining the subtype-specific differences in migratory behavior [145].

To mimic the mechanical and structural features of the mammary duct, Kulwatno et al. developed a biomimetic ductal model with tunable matrix stiffness. Using MCF10A, MCF10CA1, and MDA-MB-231 cells, they observed that while MDA-MB-231 cells invaded across all stiffness conditions, the non-tumorigenic and pre-invasive cells only exhibited protrusive activity in softer matrices. This highlights the stiffness-dependent invasive potential and utility of the model in studying DCIS-to-IDC transitions in TNBC contexts [146].

Interstitial fluid flow has been shown to be a potent driver of TNBC invasion. In a perfused microfluidic platform, Shields et al. and Y. L. Huang et al. demonstrated that autologous chemotaxis—wherein cancer cells migrate along gradients of their own secreted chemokines—is enhanced by interstitial flow. This flow alters stromal signaling, disrupts cell adhesion, and downregulates E-cadherin, thereby promoting invasion even among otherwise non-aggressive cells within co-cultured spheroids [147,148].

Finally, to dissect subtype-specific invasion, Moon et al. designed a microfluidic system with a ductal structure embedded in 3D collagen. Invasion assays revealed that TNBC lines (MDA-MB-231 and SUM-159PT) readily infiltrated the matrix, in contrast to the non-invasive MCF-7 luminal A cells. SUM-159PT cells exhibited the highest degree of matrix degradation and CD24 expression, reinforcing the aggressive nature of certain TNBC variants and validating the platform’s predictive capabilities [149].

Figure 2 highlights the range of possible readouts, from molecular and proteomic analyses to multi-organ metastasis modeling.

Although microfluidic models are becoming more precise at embodying the in vivo state, widely used breast cancer cell lines fail to capture the genetic and phenotypic heterogeneity of patient tumors, while primary cells are difficult to isolate and sustain in culture, thus limiting the long-term fidelity and translational relevance of current models. Future research should be directed towards integrating patient-derived organoids, induced pluripotent stem cell (iPSC)-derived breast cancer models, and biobanked patient-derived scaffolds, which together could better preserve intra- and intertumoral heterogeneity, maintain long-term viability, and enhance the translational fidelity of breast cancer-on-a-chip systems.

## 7. Modeling the Metastatic Process

One of the domains in which microfluidic technology provides unparalleled insight into cancer progression, compared with other oncologic modeling platforms, is the study of immune evasion and metastasis. These systems enable the recreation of a comprehensive tumor microenvironment (TME) or tumor–immune microenvironment (TIME), as well as the simulation of key metastatic steps, including intravasation, extravasation, and colonization of distant organoids or tissue constructs. Cancer-on-a-chip devices and three-dimensional microperfused platforms allow the detailed replication of histopathological features, the dynamic interactions between diverse cell types, and the impact of fluid dynamics and shear stress. Furthermore, they facilitate quantitative assessment of parameters such as trans-epithelial electrical resistance (TEER), which is critical for evaluating barrier integrity during metastatic dissemination [150].

Metastasis involves a series of at least five sequential steps: (i) **invasion**, where cancer cells breach the basement membrane and surrounding tissue to escape the primary tumor; (ii) **intravasation**, the entry of cancer cells into the bloodstream by crossing the vessel wall; (iii) **circulatory survival**, where cells persist despite mechanical and immune challenges; (iv) **extravasation**, the exit of cancer cells from the bloodstream into distant tissues; and (v) **colonization**, the growth and establishment of secondary tumors at metastatic sites [151].

Breast cancer (BC) metastasis is a complex, heterogeneous, and organotropic process, with tumor cells preferentially colonizing specific distant organs such as the lymph nodes, bone (30–60%), lungs (21–32%), liver (15–32%), and brain (4–10%) [28]. Less frequent sites like the gastrointestinal tract and adrenal glands have also been reported [152]. This non-random distribution—commonly referred to as *organotropism*—is governed by a dynamic interplay between tumor-intrinsic gene expression programs, breast cancer subtypes, circulatory patterns, and the unique microenvironment of each target organ [153,154].

The seminal “seed and soil” hypothesis by Paget (1889) remains foundational in explaining this selective dissemination, proposing that the success of metastatic colonization depends not only on the characteristics of circulating tumor cells (the “seeds”) but also on the permissiveness of the distant tissue microenvironments (the “soil”) [155]. This principle has been further refined with the discovery of *pre-metastatic niches* (PMNs)—pro-survival microenvironments primed by tumor-derived factors and extracellular vesicles prior to metastatic colonization, which facilitate immune modulation, vascular remodeling, and stromal recruitment [154].

The molecular mechanisms driving this organotropism are diverse and intricately regulated. Breast cancer cells manipulate key genes involved in angiogenesis, epithelial–mesenchymal transition (EMT), migration, invasion, cell cycle regulation, and apoptosis evasion. These processes are controlled by transcriptional regulators such as E2Fs, HIFs, SOX, ETS, and classic EMT drivers [156]. Furthermore, subtype-specific patterns of metastasis have been widely observed: for example, ER-positive luminal BC preferentially metastasizes to bone [157,158], HER2-positive BC to the liver—often with striking spatial and temporal heterogeneity [159,160]—and triple-negative breast cancer (TNBC) as well as basal-like subtypes to the brain [28,161]. Organ-specific breast cancer metastasis models and their corresponding drivers are summarized in Table 3. Recent bulk RNA-sequencing studies have deepened our understanding of this specificity, revealing that BC cells may adapt to and co-opt organ-specific gene expression profiles to facilitate metastatic survival. Genes such as CD44 and ABCC5 are elevated in bone metastases, POSTN and APOH in liver lesions, and SFTPB and SFTA2 in lung metastases—aligning with the functional roles of these genes in their respective tissues. Although CTBP1 has been suggested to influence brain metastasis, supporting evidence remains limited [162].

Microfluidic systems provide a unique opportunity to integrate BC cells with various tissue types through vasculature-mimicking pathways. This integrative approach enables the stepwise investigation of metastatic progression, from local invasion and intravasation to extravasation, seeding, and the establishment of metastatic niches. Considering the primary target organs of BC metastasis, researchers have developed a range of biomimetic devices that model interactions between metastatic cells and lymph nodes, blood vessels, bone, liver, and brain.

Recent advances in microfluidic technology and mechanical measurement have accelerated research in tumor fluid mechanics, with growing evidence highlighting fluid shear stress (FSS) as a critical factor in cancer progression and metastasis. FSS—defined as the frictional force between fluid layers in laminar flow and quantified by fluid viscosity and shear rate—regulates various cellular responses, including endothelial polarity, cytoskeletal remodeling, post-translational modifications, and gene expression [163]. During metastasis, tumor cells are exposed to both interstitial and blood shear stress. FSS promotes metastasis, lymphatic drainage, and drug transport, whereas blood shear stress plays a dual role. In certain contexts, it facilitates tumor cell adhesion and extravasation, but it can also promote circulating tumor cell (CTC) clearance and induce cell cycle arrest [164,165,166].

Within 45 h, MX-1 breast cancer cells traversed the collagen barrier and entered adjacent perfusion channels. The invading cells exhibited diverse motility patterns, including (i) amoeboid-like migration, characterized by a rounded, plastic morphology with rapid directional changes; (ii) mesenchymal-like migration, distinguished by an elongated shape and prominent leading-edge protrusions; and (iii) collective migration, in which cells preserved intercellular junctions and advanced as a coordinated group—a phenomenon previously reported predominantly in animal models [167].

### 7.1. Lymphatic Metastasis-on-a-Chip Models

Lymphatic angiogenesis and intravasation form the primary route for early dissemination in breast cancer, marking the lymphatic system as a critical conduit for metastasis. Microfluidic platforms have emerged as powerful tools to replicate and dissect this pathway with physiological relevance.

**Cho et al.** developed an integrated lymphatic vessel–tumor tissue–blood vessel microfluidic chip to recapitulate the tumor microenvironment during lymphatic metastasis. The study identified interleukin-6 (IL-6) as a critical mediator of intercellular communication. Exposure to IL-6 induced epithelial–mesenchymal transition (EMT) and markedly enhanced the invasive capacity of multiple breast cancer subtypes by up to eightfold. Moreover, IL-6 promoted VEGF secretion by human lymphatic endothelial cells, which subsequently guided the sprouting and migration of human venous endothelial cells toward the lymphatic compartment, effectively modeling pro-metastatic vascular remodeling [168].

Expanding the scope to secondary lymphatic organs, **German et al.** introduced a lymph node-on-a-chip (LNOC) system that simulates metastatic tumor formation within lymph nodes. This platform incorporated a collagen sponge scaffold embedded with 3D spheroids of 4T1 breast cancer cells, emulating the structural and porous characteristics of native lymphoid tissue. The model demonstrated potential for pharmacological testing, offering a controlled environment for evaluating anti-metastatic therapeutics in a lymph node context [169]. To capture the influence of breast cancer subtype heterogeneity on lymphatic behavior, **Ayuso et al.** developed a tumor–lymphatic microfluidic interface using MCF7 (ER+) and MDA-MB-231 (TNBC) cells co-cultured with lymphatic endothelial cells. Transcriptomic profiling revealed distinct alterations in pathways related to angiogenesis, permeability, metabolism, hypoxia, and apoptosis, depending on the cancer subtype. These molecular signatures translated into observable disruption of the endothelial barrier, validating the model’s utility in studying subtype-specific dynamics of lymphatic invasion [170].

### 7.2. Bone Metastasis-on-a-Chip Models

The bone is one of the most frequent and clinically significant sites of breast cancer metastasis, owing to its rich cellular and molecular microenvironment. Organ-on-a-chip technologies have been increasingly employed to model this complexity, including the influence of neural, vascular, and immune components.

A **bone-on-a-chip (BoC)** model enabling spontaneous formation of mineralized, collagen-rich 3D bone tissue was introduced by **Hao et al.** This platform allows co-culture with metastatic BC cells, optimizing spatial contact with the bone matrix and facilitating high-throughput observation of cancer–bone interactions [171]. Apart from the matrix itself, the perivascular niche of the bone marrow was modeled as well. **Marturano-Kruik et al.** established a triculture chip containing mesenchymal stem cells, an osteogenic matrix, and self-assembled vasculature. Under physiological shear stress and oxygen gradients, breast cancer cells entered a dormant, drug-resistant state, offering a valuable system to study tumor quiescence and resistance mechanisms in bone metastasis [172]. The immune component of the bone metastatic niche was investigated by **Crippa et al.** using a microfluidic platform incorporating stromal cells, microvessels, and fluorescently labeled MDA-MB-231 cells. Cancer-induced vascular disruption was observed, along with active neutrophil extravasation and increased cancer cell death, highlighting immune–tumor interactions within bone [173].

A vascularized tri-culture model using osteodifferentiated mesenchymal stem cells and endothelial cells recreated a bone-mimicking matrix for studying breast cancer cell extravasation. The system revealed enhanced migration of BC cells into the bone compartment, driven by CXCR2–CXCL5 signaling, underscoring chemokine-mediated tissue specificity [174].

To explore the role of the sympathetic nervous system (SNS) in bone metastasis, **Conceição et al.** developed a humanized, multi-compartment microfluidic chip simulating paracrine signaling among sympathetic neurons, osteoclasts, and bone-tropic breast cancer cells. The interaction enhanced cancer cell aggressiveness via increased secretion of pro-inflammatory cytokines such as IL-6 and MIP-1α, supporting a neuroimmune contribution to bone metastasis [175].

### 7.3. Brain Metastasis-on-a-Chip Models

Brain metastases from breast cancer are predominantly associated with basal-like and triple-negative breast cancer (TNBC) subtypes, which exhibit a distinct affinity for soft tissues such as the brain parenchyma [28,176]. A critical barrier to cerebral colonization is the blood–brain barrier (BBB)—a highly selective interface composed of tightly connected endothelial cells, basement membranes, pericytes, astrocytes covering over 90% of the endothelial surface, and microglia [28]. This multi-layered architecture restricts the extravasation of circulating tumor cells (CTCs) into the brain, posing a major obstacle to metastatic progression.

To model and investigate this process, BBB-on-a-chip platforms have been developed, replicating key structural and functional features of the neurovascular unit. For instance, a microfluidic BBB model incorporating astrocytes and brain microvascular endothelial cells, demonstrated that MDA-MB-231 TNBC cells were capable of penetrating the BBB, mimicking patterns observed in melanoma and lung cancer, whereas other cancer types showed limited or no translocation [177].

Building on this, **Westerhof et al.** explored interactions between metastatic breast cancer cells and brain-resident cells, including astrocytes and microglia, using both parental (MDA-MB-231, TNBC; JIMT1, HER2+) and brain-seeking sublines (MDA-MB-231-BR, JIMT1-BR). MDA-MB-231-BR cells demonstrated enhanced translocation across the BBB via crosstalk with astrocytes and endothelial cells. Importantly, exposure to TNBC cells induced astrocyte activation, leading to elevated secretion of inflammatory cytokines and shifts in local metabolism—suggesting that astrocytes actively modulate the brain niche to favor metastasis [28,178]

Emerging evidence also highlights the role of tumor-derived extracellular vesicles in shaping the cerebral microenvironment. **Cui et al.** showed that breast cancer-derived exosomes disrupted normal neurodevelopmental processes and induced pathological changes in brain organoids, raising the possibility that exosomes may contribute to pre-metastatic niche formation in the brain [179].

### 7.4. Lung Metastasis-on-a-Chip Models

Lung metastasis represents a significant clinical burden in breast cancer, contributing to poor prognoses and high mortality. The 5-year survival rate for patients with stage IV breast cancer involving the lungs remains low, at approximately 10.94% [180]. A growing body of evidence implicates β4 integrin, particularly when delivered via tumor-derived exosomes, in promoting breast cancer cell survival and facilitating lung-specific metastasis. Notably, loss of β4 integrin expression increases apoptosis in vivo, underscoring its importance in metastatic progression [181].

To investigate these mechanisms in a physiologically relevant context, the breast cancer lung metastasis-on-chip (BCLMOC) platform was developed. This model integrates extracellular matrix (ECM)-mimetic channels with lung-mimetic compartments seeded with normal lung fibroblasts (NLFs), allowing real-time monitoring of MCF7 cell invasion. For the first time, this system enabled co-culture of breast cancer cells with cancer-associated fibroblasts (CAFs), revealing enhanced invasiveness in the presence of both NLFs and CAFs compared to NLFs alone. Importantly, β4 integrin inhibition significantly suppressed both invasion and gap closure rates in mono- and co-culture settings (*p* < 0.0001), confirming its role in modulating breast cancer invasiveness [182]. Complementing this, **Kong et al.** developed a biomimetic microfluidic platform to assess cancer cell extravasation into lung tissue. Their device features a vascular barrier lined with HUVECs overlaid on Cultrex Basement Membrane Extract, mimicking the lung endothelium. A porous membrane separates the vascular and lung chambers, which are coated with collagen I and seeded with primary pulmonary cells derived from rat lungs. Upon introducing breast cancer cells into the vascular channel, CXCL12 chemokine gradients induced directional migration and transmigration of circulating tumor cells (CTCs), recapitulating a key step in lung metastasis. The model was validated in vivo, where metastatic success rates reached 100% for MDA-MB-231 and 80% for MCF7, aligning with their respective metastatic propensities. Furthermore, treatment with AMD3100, a CXCR4 antagonist, effectively reduced metastatic colony formation in a dose-dependent manner, confirming the platform’s applicability for therapeutic testing [183].

### 7.5. Liver Metastasis-on-a-Chip Models

The liver is a frequent and lethal site of breast cancer metastasis, particularly in triple-negative breast cancer (TNBC), yet its organ-specific microenvironment remains poorly understood. To address this, **Kim et al.** developed a 3D liver-on-a-chip model to study the role of tumor-derived extracellular vesicles (EVs) in forming pre-metastatic niches. EVs from metastatic TNBC patients induced endothelial-to-mesenchymal transition in liver sinusoidal endothelial cells (LSECs) via TGFβ1, increasing fibronectin expression and enhancing cancer cell adhesion [184].

In an interesting comparison between liver and kidney tissue, **Tian et al.** introduced a liver–kidney-on-a-chip system using precision-cut tissue slices. The platform confirmed EV organotropism, showing that breast cancer EVs preferentially targeted the liver over the kidney. Notably, AMD3100 reduced liver tropism, validating the model for therapeutic screening [185].

A more recent approach combined lineage reprogramming and LOC modeling to suppress metastasis. Using lentiviral vectors encoding six hepatic transcription factors, MDA-MB-231 TNBC cells were reprogrammed into induced hepatocyte-like cells (hiHeps). In a GelMA-EV liver-on-a-chip, reprogrammed cells exhibited reduced oncogenic activity and limited metastatic behavior under 3D liver-like conditions, supporting reprogramming as a potential anti-metastatic strategy in TNBC [186].

Future research should focus on filling today’s research gaps, resumed in Table 4, such as the development of standardized multi-organ-on-a-chip platforms with optimized scaling laws, universal culture media, and dynamic vascular flow systems, enabling physiologically relevant crosstalk between organs and thereby providing a more faithful representation of metastatic dissemination in breast cancer.

### 7.6. Circulating Tumor Cells—Liquid Biopsy

In metastatic patients where surgery is no longer a viable solution towards curative treatment, tissue biopsy may be necessary for prognostic evaluation and targeted therapy, thus putting the patients under a certain amount of surgical stress ranging from a simple fine needle biopsy/tru-cut to full on surgical intervention under general anesthesia, depending on the type of malignancy, and posing the risk of being limited by tumor heterogeneity, preventing it from fully capturing the complete molecular landscape [187]. A novel, less detrimental technique evolved in the last years in the form of liquid biopsies. A liquid biopsy is a minimally invasive method for cancer molecular profiling that analyzes tumor-derived material in blood or other body fluids. It detects cell-free DNA (cfDNA), circulating tumor DNA (ctDNA), and circulating tumor cells (CTCs) using advanced techniques such as droplet digital PCR, deep sequencing, and biomarker-based enrichment. Clinically, liquid biopsies support early detection, prognosis, monitoring tumor evolution, predicting therapy response, and identifying resistance mechanisms. While cfDNA is easier to analyze, CTCs enable functional and single-cell studies. Repeatable and less invasive than tissue biopsies, liquid biopsies are central to precision oncology, with future applications expanding to fluids like urine, CSF, and ascites, as well as exosomes and methylation markers [188].

Circulating tumor cells (CTCs) are among the most promising liquid biopsy biomarkers, as they are rare malignant cells shed from primary or metastatic tumors into the bloodstream. They provide valuable diagnostic, prognostic, and therapeutic information, reflecting tumor burden, phenotype, and genetic heterogeneity, and play a key role in metastasis. Despite their clinical potential, CTC detection remains challenging due to their scarcity in blood. Existing platforms can be broadly divided into label-free methods, which exploit physical properties such as size or density but often yield low purity, and label-dependent methods, which rely on antigen–antibody interactions for positive selection of CTCs or depletion of leukocytes. While label-free techniques enable rapid processing, label-dependent approaches offer higher specificity (with CTC recovery of 87%) but may miss CTC subpopulations lacking the chosen markers [189,190]. Microfluidic technologies overcome many of these limitations by integrating capture, enrichment, and downstream molecular analysis within a single device, enabling sensitive and efficient CTC isolation. In breast cancer, microfluidic-based liquid biopsy has emerged as a promising approach, with preclinical validation in murine blood samples and clinical studies demonstrating enhanced sensitivity for circulating tumor cell (CTC) detection, as well as predictive utility for disease progression, relapse, and treatment response, highlighting its potential role in precision oncology [189,191]. For the label-dependent method, epithelial cell adhesion molecule (EpCAM) glycoproteins are the most used CTC marker, with capturing efficiencies of 99.5% for BC cell-lines (MCF-7) [192]. Although vastly used, EpCAM detection is evaded due to ECM transition, by which stationary epithelial cancer cells transform into motile, invasive mesenchymal-like cells, driving metastasis, through which EpCAM is downregulated [193]. To address this, a novel HER2-based microfluidic device was developed for isolating CTCs from the blood of patients with HER2-expressing solid tumors such as gastric and breast cancers [194]. Dynamic characterization of both early and late-stage BC CTCs based on cytokeratin (CK8/18), epidermal growth factor (EGFR), and HER2 achieved detection of seven oncologic patients out of seven, through a nanotube-CTC-chip [195].

Zhang et al. introduced a label-free detection method, with a microfluidic device capable of both single CTCs and CTC clusters, that present higher metastatic capabilities. Moreover, they demonstrated association between CTC count and TNM stage [196]. More recently, magnetic nanoparticles (MNPs) functionalized with anti-HER2 antibodies were integrated into a microfluidic chip to isolate HER2-positive breast cancer cells. While off-chip separation reached 59.38% efficiency, the S-shaped microchannel chip improved efficiency to 96% at 0.5 mL/h, reduced analysis time by 50%, and avoided clogging—demonstrating clear potential for clinical CTC applications [197]. Liquid biopsy-compatible characterization techniques are rapidly advancing and hold strong potential for integration with breast cancer-on-a-chip platforms. Optical and imaging-based methods, such as flow cytometry, imaging flow cytometry, and nanoparticle tracking analysis, enable phenotypic profiling of circulating tumor cells (CTCs) and extracellular vesicles (EVs) with high sensitivity and specificity. Spectrochemical approaches like ATR-FTIR spectroscopy further allow multiplexed biochemical characterization of nucleic acids, proteins, lipids, and metabolites in patient-derived fluids. Complementary -omics and single-molecule technologies—including droplet digital PCR (ddPCR), targeted and whole-genome next-generation sequencing (NGS), and RNA sequencing—provide quantitative resolution of tumor-derived nucleic acids (ctDNA, cfRNA, microRNAs), facilitating detection of actionable mutations (e.g., PIK3CA, ESR1, BRCA1/2), minimal residual disease, and resistance mechanisms. Proteomic and metabolomic profiling of EV cargo adds an additional dimension for biomarker discovery [190].

Apart from the clear advantage of being able to run and evaluate CTCs directly on-chip, microfluidic devices offer options for other types of liquid biopsies as well. For example, Gwak et al. developed a microfluidic platform capable of capturing, purifying, and enriching circulating cell-free DNA (cfDNA) from the plasma of breast cancer patients, thereby minimizing sample loss and reducing processing time for genomic analysis and metastatic risk assessment [198]. Furthermore, exosomes, which carry abundant tumor-derived information, represent promising non-invasive biomarkers, but their analysis in blood is limited by complex preparation steps and low sensitivity. To overcome these challenges, a filter-electrochemical microfluidic chip (FEMC) was developed to directly detect and classify BC miRNA from whole blood without extensive purification. By targeting exosome-associated biomarkers (PMSA, EGFR, CD81, and CEA), the FEMC successfully classified BC in both mouse models and patient samples within one hour, with detection sensitivity of up to 1.0 × 10^4^ particles/mL [199]. In a similar RNA based approach, Lim et al. developed a microfluidic exosomal mRNA sensor (exoNA-sensing chip) integrating 3D nanostructured hydrogels for one-step detection of exosomal ERBB2, with robust performance in both in vitro and in vivo BC samples (high sensitivity (LOD = 58.3 fM); used minimal sample volume <100 μL) [200].

Moving away from blood samples towards an even less invasive liquid biopsy method, Mun et al. developed a herringbone-patterned nanochip coated with HER2-antibody for exosome isolation and analysis directly from BC patients’ urine samples [201].

Liquid biopsy has emerged as a transformative approach in breast cancer, offering a minimally invasive means to capture tumor heterogeneity, monitor disease dynamics, and guide precision therapies in real time. Its ability to detect circulating tumor cells, cell-free nucleic acids, and extracellular vesicles makes it a powerful alternative and complement to tissue biopsies, particularly for longitudinal monitoring and early relapse detection [190]. Importantly, the integration of **microfluidic technologies** further enhances this potential, enabling highly sensitive, rapid, and low-volume analyses with streamlined capture, enrichment, and molecular characterization on a single platform.

This technological convergence has already moved from bench to bedside: clinical trials are now actively incorporating microfluidic-based liquid biopsy devices for breast cancer diagnosis and monitoring. In one such study (NCT02948751), patients undergoing evaluation for leptomeningeal metastasis provide cerebrospinal fluid (CSF) samples, which are assessed for circulating tumor cells and cell-free DNA using the OncoCEE™ microfluidic platform—demonstrating the application of microfluidics in central nervous system metastatic surveillance. Another trial (NCT04239105) explores a microfluidic and Raman spectroscopy-based device for CTC detection in peripheral blood (see Table 5). This platform aims not only to distinguish between healthy individuals and breast cancer patients, but also to monitor chemotherapy efficacy and disease progression in real time.

Together, liquid biopsy and microfluidic innovation pave the way toward more precise, accessible, and dynamic management of breast cancer, transforming clinical decision-making and ushering in a new era of personalized oncology.

## 8. Discussion

Breast cancer remains a formidable clinical and scientific challenge, shaped by its remarkable heterogeneity, dynamic tumor microenvironment, and multifaceted metastatic behavior. Traditional in vitro and in vivo models, though historically invaluable, have reached their limits in replicating the intricate biological realities of this disease. Breast cancer-on-a-chip (BCOC) technologies have emerged as a transformative paradigm—merging the spatial resolution of microfluidics with the biological fidelity of 3D tissue modeling—to capture the hallmarks of tumorigenesis, immune evasion, drug resistance, and organ-specific metastasis with unprecedented precision.

This review has systematically explored the evolution of BCOC platforms, from their fabrication materials and techniques to their application in modeling distinct molecular subtypes, tumor–stroma–immune interactions, and multistep metastatic processes. Crucially, we highlighted how these platforms enable dynamic co-culture, simulate physiological gradients, and offer patient-specific modeling capabilities that directly respond to the unmet needs of personalized oncology.

Microfluidic modeling of ductal carcinoma in situ, luminal A breast cancer, and triple-negative subtypes has already begun to reshape our understanding of disease behavior under near-physiological conditions. Organ-specific metastasis-on-a-chip systems—ranging from bone to brain, liver, lung, and lymph nodes—offer unique insights into organotropism, dormancy, and therapy resistance. Coupled with cutting-edge liquid biopsy integration, these devices now enable real-time functional evaluation of circulating tumor cells, exosomes, and cell-free nucleic acids, transforming the landscape of non-invasive diagnostics and treatment monitoring.

While 3D breast cancer cultures in microfluidic systems provide a more physiologically relevant representation of tumor behavior, they also introduce intrinsic complexity due to dynamic biochemical gradients, heterogeneous cell populations, and evolving stromal interactions. To capture this complexity, multiple parameters—including oxygenation, extracellular acidity, cytokine release, and barrier integrity—must be monitored longitudinally and in parallel. This necessitates platforms that support parallelization, allowing simultaneous testing of multiple conditions or patient-derived samples, and in some cases reusability, which can reduce costs and increase reproducibility. Advances in integrated sensing technologies now enable high-sensitivity and high-specificity monitoring within such platforms: trans-epithelial electrical resistance (TEER) electrodes quantify vascular integrity; optical and electrochemical probes measure oxygen, pH, and metabolites; while nanoplasmonic and antibody-functionalized biosensors allow real-time detection of low-abundance breast cancer biomarkers, including HER2, EpCAM, and extracellular vesicles. Together, these innovations extend the utility of breast cancer-on-a-chip devices from structural modeling to functional, multiparametric diagnostics under near-physiological conditions.

Despite their promise, breast cancer-on-a-chip platforms also present specific risks and limitations that must be carefully acknowledged. From a technical standpoint, commonly used materials such as PDMS absorb small hydrophobic molecules, distorting pharmacokinetic readouts; stem cell–derived models often exhibit immature or fetal-like phenotypes, limiting their physiological relevance; while ECM substitutes such as Matrigel introduce batch-to-batch variability [202]. Biologically, most platforms remain simplified, lacking full immune representation or failing to capture intratumoral heterogeneity, which poses the risk of oversimplification. Clinically, the predictive validity of breast cancer-on-a-chip models is not yet fully established; there is a risk that therapeutic responses observed on-chip may not always translate into patient benefit, especially in the absence of standardized validation frameworks. Furthermore, the reuse of devices raises sterility concerns, while the handling and transport of patient-derived samples involves biosafety and ethical risks. Addressing these challenges through standardized protocols, regulatory guidance, and advanced biomaterials will be critical for the safe and effective translation of these technologies.

Breast cancer-on-a-chip platforms are increasingly demonstrating bedside potential by directly addressing the limitations of conventional preclinical models. Custom invasion/chemotaxis and extravasation chips have been shown to discriminate between metastatic phenotypes of breast cancer cells in tissue-specific microenvironments (lung, liver, breast), with invasion and extravasation patterns closely reflecting clinical metastatic data. Such systems open the possibility for improving diagnostic stratification and guiding therapeutic choice [203]. Similarly, tumor-on-a-chip models incorporating an endothelial barrier enable the transmigration and infiltration of immune cells, thereby supporting the preclinical development of immunotherapies such as CAR-T cells, including the ability to integrate patient-derived organoids for personalized testing. Furthermore, high-throughput microfluidic platforms (e.g., OrganoPlate^®^) allow simultaneous culture of dozens of perfused 3D breast cancer microtissues using limited patient-derived material, thereby enabling real-time drug sensitivity testing in a clinically feasible timeframe [204]. In parallel, liquid biopsy technologies already integrated into chip-based approaches allow non-invasive, repeatable monitoring of circulating tumor cells, exosomes, and cell-free nucleic acids. Their translational potential is underscored by two ongoing phase III clinical trials investigating the direct bedside application of liquid biopsies for breast cancer management. Collectively, these examples highlight that breast cancer-on-a-chip systems are not only valuable research tools but are progressing toward clinically relevant, personalized applications for diagnosis, therapy selection, and monitoring.

Despite rapid advances, breast cancer-on-a-chip technologies still face several important challenges. From a fabrication standpoint, material limitations such as PDMS absorption of hydrophobic drugs can distort pharmacokinetic readouts, while glass and thermoplastic alternatives may be costly or lack gas permeability. Microfabrication techniques including soft lithography, injection molding, and 3D printing offer rapid prototyping but often result in low throughput, high costs, or geometrical uncertainties that compromise reproducibility. Long-term cultures are further hindered by risks of channel clogging, bubble formation, and instability of hydrogel-based scaffolds, limiting assay duration and standardization [205]. Sustaining long-term, physiologically relevant 3D cultures is technically demanding, and modeling the full complexity of the tumor microenvironment—including immune, adipose, and myoepithelial compartments—remains incomplete. Incorporating patient-derived material raises challenges of scalability, cell availability, and phenotypic stability. Furthermore, regulatory acceptance is still lacking, as no harmonized validation frameworks exist to benchmark organ-on-a-chip against established preclinical models. Addressing these challenges will be essential to translate breast cancer-on-a-chip systems from experimental tools to robust clinical platforms.

Importantly, international regulatory bodies such as the U.S. Food and Drug Administration (FDA) and the European Medicines Agency (EMA) have introduced new policies aimed at reducing reliance on animal models for drug development and disease research [206]. These shifts further propel the scientific community toward embracing more ethical, cost-effective, and physiologically relevant platforms.

A key strength of breast cancer-on-a-chip platforms is their capacity to recapitulate aspects of the disease that are challenging to reproduce using conventional systems. These include the complex architecture of the tumor microenvironment, dynamic interactions between cancer and immune cells, and the ability to assess drug responses under physiologically relevant flow conditions. By leveraging these unique features, breast cancer-on-a-chip technology generates insights into disease progression and therapy resistance that exceed those provided by traditional in vitro or animal models, underscoring its translational relevance for oncological research and drug development.

## 9. Conclusions

Despite current limitations in scalability and standardization, breast cancer-on-a-chip technologies are progressing steadily. The integration of microengineering, tumor biology, and computational approaches is expected to support the development of more complex systems, including multi-organ models and predictive analytics for therapy selection. Although the concept of tailoring treatment to patient-derived tumors cultured on a chip remains under investigation, recent advances suggest that such strategies may become clinically feasible in the near future. Breast cancer-on-a-chip platforms hold strong potential to enhance our understanding of disease mechanisms, improve drug testing, and contribute to more individualized therapeutic strategies, while simultaneously supporting the long-term goal of reducing and ultimately replacing animal models in cancer research.

## Figures and Tables

**Figure 1 cancers-17-03160-f001:**
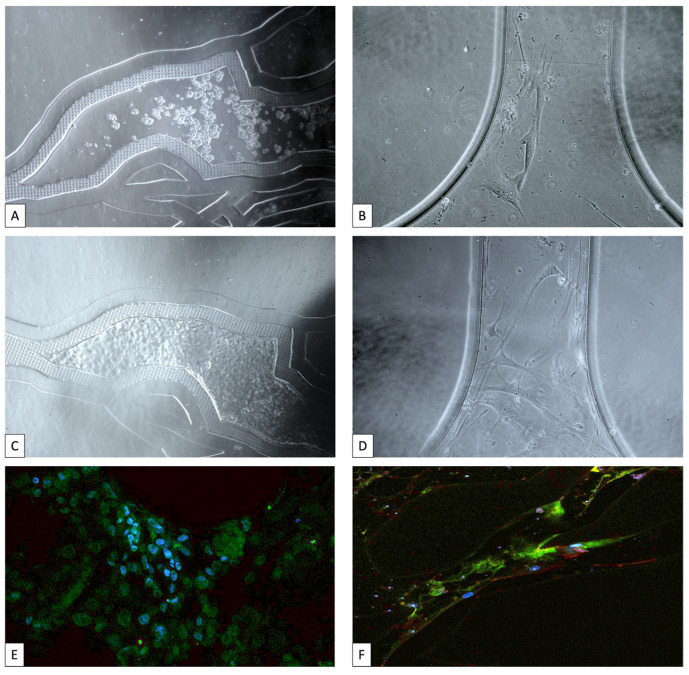
Co-culture of MCF-7 cell lines (**A**,**B**,**E**) with hBM-MSCs on a microfluidic perfused platform. Note the dynamic MCF-7 cells-derived tumor development on days 7 (**A**) and 28 (**C**) and the positivity of E-cadherin (green, (**E**)) and hBM-MSCs differentiation on days 7 (**B**) and day 28 (**D**), followed by the acquisition of VE-cadherin ((**F**), green) with the continued expression of (**F**) actinin (patches of red immunofluorescence).

**Figure 2 cancers-17-03160-f002:**
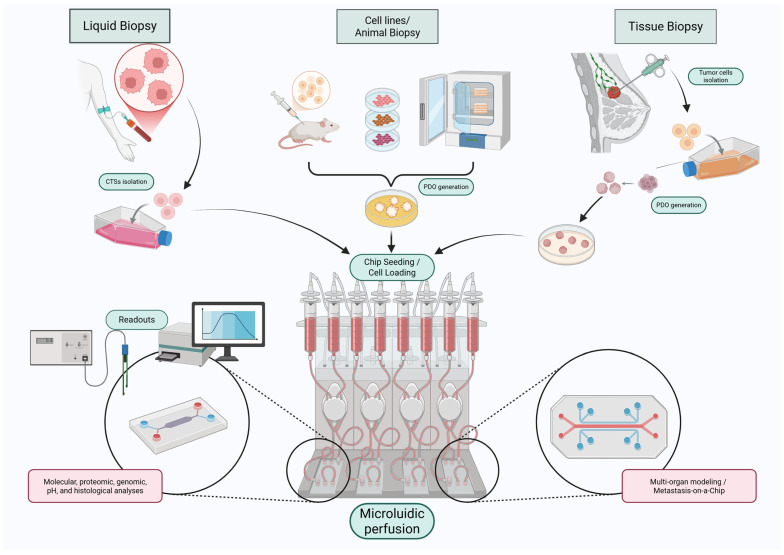
From biopsy to microfluidics: modeling tumor biology on a chip. This diagram shows the development of tumor-on-a-chip platforms using cells from liquid biopsy (CTCs), tissue biopsy, or animal models. Isolated cells are expanded (e.g., organoids) and seeded into microfluidic chips with controlled perfusion. The system supports downstream analysis via TEER, live-cell fluorescence imaging, molecular profiling, pH monitoring, and histology, as well as advanced applications such as metastasis-on-a-chip and multi-organ integration for personalized cancer modeling.

**Table 1 cancers-17-03160-t001:** Limitations of conventional models vs. advantages of breast cancer-on-a-chip platforms.

Model Type	Key Limitations	Breast Cancer-on-a-Chip Solutions
**2D In Vitro Cultures**	- Lack of 3D architecture - Absence of stromal/immune cell interactions - No physiological gradients	- 3D microenvironments with co-cultures (tumor, stromal, immune cells) - Integrated gradient systems
	- Homogeneous cell populations	- Incorporation of patient-derived heterogeneous cells
**Animal Models**	- High cost and long duration - Species-specific differences in drug response	- Human-cell-based systems with better translational relevance
	- Ethical concerns - Limited ability to model patient-specific tumors	- Personalized chips from patient-derived cells
**Static 3D Models**	- No dynamic flow or mechanical cues - Limited control of microenvironmental factors	- Perusable channels with dynamic control over shear stress, oxygen, and nutrients
	- Low scalability for high-throughput screening	- Scalable platforms suitable for drug screening

**Table 2 cancers-17-03160-t002:** Organ/cancer-on-a-chip fabrication technique.

Technique	Advantages	Limitations	Typical Materials
**Soft** **Lithography**	High-resolution, biocompatible, cleanroom-free	Needs master mold	PDMS, polystyrene
**Photolithography**	High precision and batch fabrication	Cleanroom required, expensive	Glass, silicon
**Injection** **Molding**	Scalable, reproducible	High setup cost, inflexible design	PMMA, COC
**Hot Embossing**	Good replication fidelity	Heat-sensitive, slow cycles	PC, PLA
**Etching**	Sub-micron resolution	Toxic reagents, complex process	Glass, silk fibroin
**Laser Cutting**	Rapid prototyping, low-cost	Thermal damage, toxic residues	Paper, PET
**3D Printing**	Customizable, broad material range	Size resolution, slow throughput	PEG, collagen, alginate
**Electrospinning**	Biomimetic ECM, tunable fibers	Requires integration with microfluidics	PCL, PLA, gelatin

**Table 3 cancers-17-03160-t003:** Target organ metastasis.

Target Organ	Preferred Subtypes	Metastasis Drivers	OoC Features Modeled
Bone	ER+ luminal	CXCL12–CXCR4 axis, dormancy mechanisms	Triculture chips, shear stress, vascular mimicry
Brain	TNBC, HER2+	BBB disruption, astrocyte signaling	BBB-on-a-chip, astrocyte/EC co-cultures
Liver	TNBC	EVs, fibronectin expression	LOC with induced hepatocytes, EV signaling
Lung	TNBC, luminal B	β4 integrin, CXCL12 gradient	Lung-on-chip, HUVEC layers, exosome tracking
Lymph Nodes	All	IL-6 signaling, VEGF-mediated remodeling	LN-on-chip, lymphatic EC co-culture

**Table 4 cancers-17-03160-t004:** Research gaps and future solution-based analysis.

Research Gap.	Description	Future Need
**Lack of Standardization** **and Reproducibility**	No harmonized protocols for chip design, ECM, or readouts; hinders reproducibility.	Establish standardized fabrication and validation protocols.
**Incomplete TME** **Representation**	Key cell types (myoepithelial cells, adipocytes, lymphoid aggregates) often excluded.	Integrate full immune–stromal– adipose–myoepithelial complexity.
**Limited Modeling of** **Tumor Heterogeneity**	BCOC rarely captures polyclonality or spatial tumor heterogeneity.	Design chips with spatial/temporal heterogeneity and clonal tracking.
**Short-Term Culture** **Limitations**	Most platforms limited to short durations (<1 week), impeding chronic drug modeling.	Develop long-term perfused models with treatment simulation capabilities.
**Single-Organ Metastasis** **Modeling**	Few models recreate multi-organ metastatic cascades or pre-metastatic niche formation.	Connect multi-organ systems with real-time tracking of tumor migration.
**Neglect of Biomechanical Forces**	Models lack simulation of stiffness, compression, and tissue deformation.	Incorporate mechanical strain, pressure, and tension cues.
**Low Scalability for** **Personalized Testing**	Platforms are low-throughput, unsuitable for real-time therapeutic screening.	Miniaturize for multiplexed patient-on-a-chip drug screening.
**Unclear Regulatory** **Integration**	Despite regulatory interest, there is no defined validation path for BCOC in preclinical pipelines.	Define regulatory roadmaps and align with FDA/EMA guidelines.

**Table 5 cancers-17-03160-t005:** Clinical trials of microfluidic liquid biopsies.

Trial ID	Application	Technology	Target Analyte
NCT02948751	Leptomeningeal metastasis detection	OncoCEE™ microfluidics	CSF-derived CTCs and cfDNA
NCT04239105	CTC detection, therapy monitoring	Microfluidics + Raman spectroscopy	Peripheral blood CTCs

## Data Availability

No new data were created or analyzed in this study. Data sharing is not applicable to this article.

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
