# Peer review of "Microengineered Breast Cancer Models: Shaping the Future of Personalized Oncology"

_cancers, 2025, doi:10.3390/cancers17193160_

Round 1

Reviewer 1 Report

Comments and Suggestions for Authors

The review submitted to cancers j by Popoiu et al, 2025, titled “Breast Cancer-on-a-Chip: Bridging Complexity, Precision, and the Future of Personalized Oncology

  1. The title reflects the idea, but better to be changed to “"Microengineered Breast Cancer Models: Shaping the Future of Personalized Oncology"
  2. The abstract is better to be structured
  3. Start the introduction with the subheading background.
  4. The paper conclusion is long and more like a comprehension
  5. This would be better a systematic review metanalysis
  6. What about the clinical applicability
  7. What about the bed side?
  8. Challenges need to be stated
  9. The study questions and the aim are not well defined.
  10. Figure 1 is comprehensive
  11. The is a risk of bias ROB but the authors comprehend this method as a review not applicable
  12. Rationalize why this is not a metaanalysis study
  13. List of abbreviations are provided
  14. Conclusion is over rated needs to be rewritten
  15. Future prospectives and recommendations need to be added as a separate subheading
  16. The manuscript is “Overloaded with buzzwords” that is overpromising
  17. What about risks, they are generic
  18. There is lack of spcificity
  19. Phrases like “the Future of Personalized Oncology” imply a visionary or review paper, not necessarily original experimental research.
  20. Narrow focus → highlight what’s unique (e.g., tumor microenvironment, immune-tumor interactions, drug testing).
  21. Is this a review or perspective?
  22. Data-driven research (technical, precise) are lacking please add them

Author Response

Response to Reviewer 1

Dear Reviewer 1,

We sincerely thank you for the time and effort you invested in evaluating our manuscript. Your insightful comments and constructive suggestions were carefully considered and have helped us improve the quality and clarity of the paper.

Below we provide a point-by-point response. All corresponding changes in the manuscript have been highlighted in red.

With our gratitude,
On behalf of all authors,
Anca Maria Cimpean and Tudor Popoiu

1. Title reflects the idea, but better to be changed to “Microengineered Breast Cancer Models: Shaping the Future of Personalized Oncology.”
Thank you for the suggestion. We prefer to retain the term “breast cancer-on-a-chip” rather than “microengineered” to preserve consistency with the terminology used in the field. However, we have shortened the title accordingly.

2. The abstract is better to be structured.
We have restructured the abstract to improve clarity and readability. Thank you for the suggestion (lines 45-69).

3. Start the introduction with the subheading background.
We revised the Introduction and included two subheadings to provide a clearer structure, including a dedicated “Background” section (lines 73-133).

4. The paper conclusion is long and more like a comprehension.
Thank you for this remark. We addressed it by creating a separate, concise Discussion section and leaving a shorter, focused Conclusion at the end (lines 1009-1122).

5. This would be better as a systematic review or meta-analysis.
We appreciate the suggestion. However, given the diversity of the topic, we believe that a meta-analysis would not be feasible. A narrative review is more appropriate, as it allows us to cover different key themes, provide context, and explore emerging concepts. Moreover, the scope of our work aligns more closely with the definition of a narrative review, which emphasizes flexibility and interpretation.

6. What about the clinical applicability?
Clinical applicability remains the long-term goal of this field. We briefly discussed this aspect and included examples of ongoing clinical trials. If specific relevant studies were overlooked, we would be grateful to include them upon your suggestion.

7. What about the bedside?

We thank the reviewer for this helpful comment. We have now expanded the Discussion to specifically highlight the translational and bedside potential of breast cancer-on-a-chip technologies. In the revised version, we discuss: (i) invasion/chemotaxis and extravasation chips that replicate organ-specific metastatic patterns in agreement with clinical data; (ii) tumor-on-a-chip models integrating endothelial barriers and patient-derived organoids to assess CAR-T cell efficacy; (iii) high-throughput platforms (e.g., OrganoPlate®) that enable parallel culture of perfused 3D breast cancer tissues for real-time drug screening using patient samples; and (iv) the integration of liquid biopsy approaches, which are currently under evaluation in two phase III clinical trials, underscoring their near-term clinical relevance (lines 1068-1081).

8. Challenges need to be stated.
We have added a dedicated subsection on challenges within the Discussion chapter (lines 1082-1097).

9. The study questions and aim are not well defined.
We clarified and explicitly highlighted the study questions and aims in both the Abstract and Introduction (marked in red). Additionally, Table 1 at the end of the Introduction provides a comparative overview that further illustrates the scope and aim of this review (lines 130-133).

10. Figure 1 is comprehensive.
Thank you for your kind appreciation.

11. There is a risk of bias (ROB), but the authors comprehend this method as a review not applicable.
We acknowledge this comment. Given that our manuscript is a narrative review, a formal ROB analysis was not applicable.

12. Rationalize why this is not a meta-analysis study.
We addressed this point in our response to comment 5.

13. List of abbreviations are provided.
Thank you for noting this.

14. Conclusion is overrated and needs to be rewritten.
We have revised and toned down the Conclusion to make it more balanced and objective (lines 1112-1122).

15. Future perspectives and recommendations need to be added as a separate subheading.
We thank the reviewer for this constructive suggestion. While we agree that future perspectives and recommendations are valuable, we felt that adding a separate section would unnecessarily lengthen the review. Instead, we expanded the Discussion to highlight key future directions and challenges, ensuring the review remains concise while still addressing this important point.

16. The manuscript is “overloaded with buzzwords” and overpromising.
Thank you for pointing this out. We carefully revised the manuscript and toned down language that could be considered exaggerated.

17. What about risks, they are generic.
We have expanded this section in the Discussion, providing a more detailed description of risks specific to breast cancer-on-a-chip applications (lines 1047-1061).

18. There is lack of specificity.
We thank the reviewer for this valuable feedback. We revised the text to include more quantitative performance data (e.g., capture efficiency, detection limits, culture duration), examples of compatible characterization techniques (optical, imaging, omics), and translational applications from recent studies. We also expanded the discussion of risks and challenges. Together, these changes enhance the specificity of the review.

19. Phrases like “the Future of Personalized Oncology” imply a visionary paper, not necessarily original research.
We revised the phrasing to better reflect the narrative review nature of the manuscript.

20. Narrow focus → highlight what’s unique (e.g., tumor microenvironment, immune–tumor interactions, drug testing).
Thank you for this helpful suggestion. We added a paragraph in the Conclusion emphasizing the unique strengths of breast cancer-on-a-chip platforms, including their ability to replicate the tumor microenvironment, model immune–tumor interactions, and evaluate drug responses under physiologically relevant flow.

21. Is this a review or a perspective?
As mentioned in response to comment 5, this is a narrative review.

22. Data-driven research (technical, precise) is lacking; please add.
We added quantitative data on relevant techniques and complemented the text with an illustrative figure from our own experience (Figure 1) (lines 568-576; 926; 976-978; 981-982).

Reviewer 2 Report

Comments and Suggestions for Authors

The Review paper “Breast Cancer-on-a-Chip: Bridging Complexity, Precision, and the Future of Personalized Oncology” by T.A. Popoiu et al. discusses the state-of-the-art of emerging breast-cancer on-chip technologies. This Review summarizes the latest advancements in the development of devices capable to integrate 3D cell cultures while mimicking the tumor microenvironment near-physiological conditions to provide superior microfluidic modeling of breast cancer pathology and its subtypes. I would like the authors to address the following points:

  • The materials and fabrication techniques used to produce these on-chip platforms for breast cancer screening have been extensively explained. On the other side, I believe that the authors do not provide sufficient information on the characterization techniques based on liquid biopsy (optical and imaging techniques, -omics technologies for single-molecule sensing, etc.) that are compatible with these devices and that have been used to detect and monitor over time breast cancer cells and related biomarkers, such as proteins and extracellular vesicles. This aspect should be discussed in more detail in the Review.
  • Having access to 3D cell cultures in near-physiological conditions has the potential to provide a better insight into the disease but the samples to test are intrinsically more complicated. This implies that several factors should be monitored over time and in parallel to gain a better insight of the disease pathology from the analyzed samples. On one hand this requires platforms capable of parallelization, and possibly reusability, while on the other side the use of sensing techniques capable of high sensitivity and high specificity is also needed. Please mention these aspects.
  • Overall, several examples of breast cancer on-chip platforms are mentioned; however, I believe that quantitative information about the performance of these devices should be mentioned, when possible, to help the reader to proper compare between them.

Author Response

Dear Reviewer 2,

We sincerely thank you for your careful reading of our manuscript and for your constructive suggestions. Your feedback has helped us to strengthen the review and make it more informative for the reader. Below, we address each point in detail.

1. Lack of sufficient information on characterization techniques compatible with breast cancer-on-a-chip platforms (optical, imaging, -omics, etc.)
We thank the reviewer for highlighting this important point. In response, we have expanded the manuscript to include a more detailed overview of liquid biopsy-compatible characterization techniques. Specifically, we discuss optical and imaging-based approaches such as flow cytometry, imaging flow cytometry, and nanoparticle tracking analysis for CTC and extracellular vesicle (EV) profiling, as well as spectrochemical techniques like ATR-FTIR spectroscopy for multiplexed biochemical analysis. We also highlight -omics and single-molecule technologies—including ddPCR, targeted and whole-genome NGS, RNA sequencing, and proteomics—that allow quantitative detection of circulating tumor DNA, cfRNA, microRNAs, and exosomal cargo. These tools provide high sensitivity and specificity for monitoring dynamic disease states, detecting actionable mutations, and characterizing resistance mechanisms. The inclusion of this information strengthens the translational scope of the review and clarifies how liquid biopsy readouts can be integrated with breast cancer-on-a-chip platforms (Lines 952-935).

2. Multiparametric monitoring, parallelization, and reusability of platforms
We fully agree with the reviewer’s observation that the intrinsic complexity of 3D cell cultures requires multiparametric monitoring over time to capture disease dynamics. To address this, we have expanded the Discussion to emphasize:

  • the importance of parallelized chip arrays for simultaneous testing of multiple conditions or patient-derived samples;

  • the potential for device reusability through modular thermoplastic or sterilizable designs, which enhance reproducibility and reduce costs; and

  • the integration of high-sensitivity, high-specificity sensing modalities such as TEER electrodes, optical/electrochemical probes for oxygen and pH monitoring, and antibody-functionalized or nanoplasmonic biosensors for biomarker detection.

  • lines 1029-1043

3. Lack of quantitative performance data of breast cancer-on-a-chip devices
We appreciate this remark. To improve comparability for the reader, we have incorporated quantitative performance data into the revised manuscript, including parameters such as capture efficiency, detection limits, culture duration, and sensitivity/specificity where available (lines 926; 977-978; 981-982).

Round 2

Reviewer 1 Report

Comments and Suggestions for Authors

To be accepted per the manuscript is improved after addressing all comments raised.